# AI-Based Detection of Aspiration for Video-Endoscopy with Visual Aids in Meaningful Frames to Interpret the Model Outcome

**DOI:** 10.3390/s22239468

**Published:** 2022-12-04

**Authors:** Jürgen Konradi, Milla Zajber, Ulrich Betz, Philipp Drees, Annika Gerken, Hans Meine

**Affiliations:** 1Institute of Physical Therapy, Prevention and Rehabilitation, University Medical Center of the Johannes Gutenberg-University Mainz, 55131 Mainz, Germany; 2Department for Health Care & Nursing, Catholic University of Applied Sciences, 55122 Mainz, Germany; 3Department of Orthopedics and Trauma Surgery, University Medical Center of the Johannes Gutenberg-University Mainz, 55131 Mainz, Germany; 4Fraunhofer Institute for Digital Medicine MEVIS, 28359 Bremen, Germany

**Keywords:** XAI, segmentation, detection, aspiration, glottis, vocal cords, endoscopy, FEES, interpretability, meaningful sequences, key frames

## Abstract

Disorders of swallowing often lead to pneumonia when material enters the airways (aspiration). Flexible Endoscopic Evaluation of Swallowing (FEES) plays a key role in the diagnostics of aspiration but is prone to human errors. An AI-based tool could facilitate this process. Recent non-endoscopic/non-radiologic attempts to detect aspiration using machine-learning approaches have led to unsatisfying accuracy and show black-box characteristics. Hence, for clinical users it is difficult to trust in these model decisions. Our aim is to introduce an explainable artificial intelligence (XAI) approach to detect aspiration in FEES. Our approach is to teach the AI about the relevant anatomical structures, such as the vocal cords and the glottis, based on 92 annotated FEES videos. Simultaneously, it is trained to detect boluses that pass the glottis and become aspirated. During testing, the AI successfully recognized the glottis and the vocal cords but could not yet achieve satisfying aspiration detection quality. While detection performance must be optimized, our architecture results in a final model that explains its assessment by locating meaningful frames with relevant aspiration events and by highlighting suspected boluses. In contrast to comparable AI tools, our framework is verifiable and interpretable and, therefore, accountable for clinical users.

## 1. Introduction

Machine learning has a huge impact on biomedical applications and will play a continuously increasing role in diagnostics and patient care [1]. The underlying AI models can be divided into two classes: white-box models and black-box models. White-box models, e.g., decision trees based on comprehensible input variables, allow the basic understanding of their algorithmic relationships; they are thus self-explanatory with regard to their mechanisms of action and the decisions they make. With black-box models, such as deep neural networks that have recently redefined the state of the art in many applications, it is generally no longer possible to understand their inner workings [2]. Instead, there are methods for the explanation of single decisions (local explainability) or attempts at deriving descriptions of specific input patterns that a trained model looks out for. Depending on the specific requirements, it is possible to apply established explanation tools, e.g., LIME, SHAP, Integrated Gradients, LRP, DeepLift or GradCAM [3]. However, even these tools require expert knowledge for the interpretation of their output, and only a few of them provide intuitively understandable decision explanations (e.g., saliency maps, prototypes or surrogate models, or contrastive and counterfactual explanations) [4]. This means that the importance of explanatory strategies will continue to increase in the future, while they are already an essential component of many AI applications today. The importance of explainability varies greatly, depending on the field [5], with the healthcare sector being one of the most demanding ones. To serve this need, technical and non-technical challenges need to be overcome. This can lead to new and further developments of suitable “hybrid” approaches that combine data- and knowledge-driven concepts and/or white- and black-box modeling attempts [6]. Additionally, behavioral or cognitive science aspects for explainable AI should be considered, such as transparency and the measurability of the explanation, as well as automated explanation adaptations for users. Employing this human–computer interaction (HCI) provides transparency to users, allowing them to trust the machine [7]. For instance, regarding digital applications that are based on video recordings, the identification of meaningful frames or key frames [8,9] in video sequences is one saliency-map approach that can be very helpful to interpret algorithmic decisions. As an example of such a perceptive human-based interpretation approach [3], we introduce a concept that can be used to facilitate the clinical diagnosis of swallowing disorders based on video–endoscopic swallowing examinations.

Disorders of swallowing are a relevant problem across various etiologies and all sectors of healthcare provision. Each year, approximately one in 25 adults will experience a swallowing problem in the United States. Dysphagia cuts across so many diseases and age groups that its true prevalence in adult populations is not fully known and is often underestimated [10]. A recent systematic review demonstrated that the presence of oropharyngeal dysphagia significantly increases healthcare utilization and cost, highlighting the need to recognize oropharyngeal dysphagia as an important contributor to pressure on healthcare systems [11]. The leading cause for the complications of dysphagia is the aspiration of boluses and saliva (i.e., materials that pass the vocal cords and enter the airways). A comprehensive review summarized that 43–54% of all acute stroke patients suffer from dysphagia and about 37% of those patients develop aspiration pneumonia, of which 3.8% die if no dysphagia diagnosis and therapy take place. The aspiration pneumonia rate in the first 14 days can be lowered from 8.2% to 1.3% (a relative risk reduction of 84%) by early screening, instrumental diagnostics, and subsequent dysphagia therapy [12].

At present, there are two instrumental diagnostics that can be regarded as gold standards: Videofluoroscopic Swallowing Study (VFSS) and Flexible Endoscopic Evaluation of Swallowing (FEES). In contrast to VFSS, FEES is appropriate for bedside administration, is radiation-free, and can be administered by speech and language pathologists and, therefore, does not rely on medical personnel, which altogether leads to far lower costs for FEES [13,14]. All these aspects limit the clinical use of VFSS. Consequently, FEES is currently the most commonly used tool for instrumental dysphagia diagnostics. With the goal of improving and systematizing training, a multi-level training curriculum was developed [15] that is now implemented within the European Society for Swallowing Disorders (ESSD). Hence, FEES is in widespread use across Europe. In 2010 in Germany, FEES was incorporated in the German version of the International Classification of Procedures in Medicine with a cost estimation of EUR 200 [16], which is lower than the retrospectively calculated mean reimbursements of USD 321.23 in the US [14]. At present, and to be in accordance with the reimbursement procedure (OPS), FEES must be performed by two persons. According to relevant literature, the duration of FEES administration varies between 30 min and 40 min [17] but can easily reach 90–120 min (authors’ own experience of >1500 FEES). Furthermore, in a very time-consuming process, data need be stored and inspected again for the diagnostic report to establish better reliability in the detection of aspiration (Krippendorff’s alpha ~0.78 vs. second video inspection frame by frame ~0.87) [18]. Beyond binary diagnosis (aspiration/no aspiration), more detailed scales, such as the Penetration–Aspiration Scale (PAS) [19], can be used to describe the results. The PAS classifies from score 1 (material does not enter the airway), via scores 2–5 (penetration of material into the larynx at different depths and with different abilities of clearance), to scores 6–8 (aspiration with different abilities for clearance, with score 8 meaning no attempt for clearance at all). In these cases, the overall inter-rater reliability (IRR) across clinicians is stated to be between 0.35 (PAS score 5), 0.56 (PAS score 7) and 0.73. (PAS score 8) [20]; PAS scores 7 and 8 are, especially, highly relevant, as they indicate aspiration. Most striking are the differences between intra-RR (0.60) and inter-RR (0.29) before specific trainings [21], but overall inter-RR scores, irrespective of clinical experience, can also reach 0.85 [22]. Hence, the more differentiated the diagnostics should be and the less the staff are trained, the less reproducible human decisions become. The low intra- and inter-RR values, especially for PAS scores 7 and 8, clearly show the existence of relevant missing rates.

Taken together, there is room for improvements in FEES in the areas of validity, reliability, and duration in the context of aspiration detection, the reduction in staff needed for administration, and the report of findings, as well as in general costs. One recent approach that addresses the problem of human misses for penetration and aspiration detection is narrow-band imaging. It is implemented in certain types of endoscopes, can be used to sharpen the optical contrasts, and has proven to increase the IRR [23,24]. However, all of the described areas for improvement could be addressed when the administration of FEES is combined with the help of an Artificial Intelligence (AI) tool that is capable of giving reproducible and quantitative output based on a frame-by-frame analysis without concentration errors.

Although no one, to date, has developed an AI tool to detect aspiration for FEES videos, various other attempts using machine-learning approaches to detect aspiration or signs of unsafe swallowing have been performed. The only high potential application is a CNN for aspiration detection of VFSS videos, with an accuracy of AUC of 1.00 [25], but, as described above, VFSS is limited in its clinical use. Further studies investigated the possibility of identifying dysphagia by means of localization of the hyoid bone or hyoid bone movements by an AI tool: on the one hand, the detection of auscultations, swallowing sounds, and vibrations is used [26,27,28,29], and on the other hand, video material (VFSS or ultrasound) [30,31,32] is used. Both approaches yield good results. In general, the detection of (impaired) swallowing based on auscultation is the subject of research in many studies [33,34,35]. Furthermore, there are studies investigating the combination of pressure build-up (lingual/palatal/pharyngeal) in combination with AI [36,37], sometimes with the additional combination of VFSS data [38]; again, promising results can be obtained [39]. Combinations of various biometric data are also used for AI-based dysphagia diagnosis [40]. Studies looking at aspiration detection using image data (VFSS) [25] or swallow-onset detection [41] also yield promising results. A different approach is an image analysis of the external neck appearance for the detection of sarcopenic dysphagia [42]. Finally, speech recordings have also been investigated for the presence of dysphagia [43]. While some of these approaches already show sufficient accuracy, these existing machine-learning models all show black-box characteristics and lack transparency regarding their classification results: After the model’s outcome, there is no gold standard (ground truth) that can be used by the examiner to validate the model’s assumptions, as they do not provide the examiner with any explanatory insight about the airways. Hence, these models only classify between healthy and at-risk for aspiration. Only a subsequent FEES or VFSS could validate the model’s outcome. Hence, for clinical users it is difficult to trust in these models and their decisions. Furthermore, such lack of transparency might not comply with requirements of the European General Data Protection Regulation (GDPR), as it prohibits decisions that are based solely on automated processing [44] and, therefore, limits practical applications in the clinical context [45].

Thus, our aim is to introduce an explainable artificial intelligence (XAI) approach to detect aspiration (i.e., of liquids, jelly, or saliva) during FEES for patients suffering from dysphagia. The automatic detection should improve IRR but also be interpretable, increasing its trustworthiness and transparency. To facilitate the detection of bolus aspiration, while at the same time achieving explainability goals, the AI tool should also learn the segmentation of relevant anatomical structures, such as the vocal cords and the glottis, a task that has previously been shown to be feasible [46,47,48]. Simultaneously, the AI tool will be trained to detect boluses that pass the glottis and become aspirated into the airways. This interpretable architecture results in a final model that explains its assessment by locating specific video frames with relevant aspiration events and by highlighting the glottis, vocal cords, and suspected boluses in situ as visual aids in meaningful frames.

## 2. Materials and Methods

### 2.1. Video Data and Annotation

Ninety-two patient videos (50 showing aspiration, 32 showing penetration, and 10 without aspiration) based on established PAS scores [22]—8-6 for aspiration, 5-2 for penetration (no aspiration), and 1 for healthy—from an already existing data set of ~1500 FEES recordings were retrospectively analyzed by two FEES experts as a basis for annotation. All recordings were made by the same type of endoscope (Orlvision, Video Rhino Laryngoscope RS1, 3.9 mm diameter, 130°/130° probe control, 90° viewing angle, 291.000 px resolution, Orlvision GmbH) and recorded on an rpSzene system (Rehder/PartnerGmbH). The study was approved by the responsible ethics committee of the State Chamber of Physicians of Rhineland-Palatinate (No.: 2021-16141-retrospektiv) and is registered with WHO (INT: DRKS00026822). We split the videos into disjunct sets for training, validation during training, and final testing. The videos were graphically annotated using the highly customizable annotation tool for data curation and quality control, SATORI, see Figure 1 [49]. To ensure a human-in-the-loop approach, the two domain experts performed the annotation. The structures of the vocal cords, the glottis (open, closed, obscured) as the region of interest (ROI), and cases of aspiration (saliva, liquid, slurry) as well as no aspiration were drawn into certain frames and served as the gold standard (ground truth) for the AI segmentation. In addition, frames not showing any of the structures or cases of aspiration were labeled as such using a frame-labeling tool, to reduce the number of false-positive segmentations when processing a full video.

These labelled pixels created the data basis for a subsequent convolutional neural network (CNN; U-Net) specifically designed for segmentation tasks. In addition to aspiration, the AI tool was trained to segment the ROI and the vocal cords, because they are easier to detect and the aspiration always appears within this region of interest.

Furthermore, the training data was augmented using geometric transformations and color modifications (rotations, zooming up to ×1.5, mirroring left-right, change of contrast, and picture brightness based on a frame mean of ±25%) to add as many variants of the annotated frames as possible. This was done in order to make the detection of aspiration not dependent on incidental features, such as sharpness or contrast, as the detection of findings becomes more robust to such distortions and shape changes when sufficiently trained with appropriate data. Such augmentation techniques are commonly used to teach modern AI models so that different positions, lighting conditions and camera angles, partial occlusions, or horizontal and vertical shifts do not represent anomalies [50]. This approach increases robustness, reducing the expected performance drop when applying the resulting model to external test data. In summary, the goal was to train the model on a sample of videos that reasonably covered the expected variability occurring during practical application. Therefore, because the quality of the FEES videos also varies in reality, videos on which the structures were rather poorly visible were also selected.

### 2.2. Deep Neural Networks for Segmentation

A 2D U-Net was chosen as neural network architecture to segment the glottis ROI, vocal cords and aspirated boluses. U-Nets were developed specifically for the segmentation of biomedical images [51] and variants of this idea have become the most commonly used and most successful architecture to date [52]. The architecture is based on a performant fully convolutional design and consists of an encoder and a decoder that produce a result at the same resolution as the input image, with skip connections that facilitate information flow and feature re-use for a detailed result. Unlike the original U-Net architecture, convolutions with zero-padding and only 32 base filters in the first convolutional layer were used. For regularization, dropout [53] and batch normalization [54] were added and PReLU [55] was chosen as the activation function. The training was carried out on videos downscaled by a factor of two to remove comb artifacts from interlaced recording, on patches of size 352 × 288 pixels and batch size 16. The training error was optimized using the Adam optimizer [56] and Dice loss function [57] with an initial learning rate of 10^−4^. The patches were sampled so that 80% included the glottis ROI and 25% of these showed aspirations, and the remainder were frames labeled as not containing the ROI. Every 500 iterations, the U-Net was evaluated on the validation data and the Jaccard score to the reference segmentation was computed. After 15 validation steps without an improvement in the Jaccard score, the training was stopped. The network state with the highest validation Jaccard score was retained and selected as the output model (“early stopping”), which helped to prevent overfitting, which this task on such a relatively small dataset (particularly when considering frames from the same video to be correlated) [47]. The model output was post-processed by selecting the largest connected component for the glottis structure and restricting the vocal cords and aspiration segmentation to this ROI.

### 2.3. Evaluation

Comparison of the overlap of the surface area (pixels) between human assignment during annotation and AI-based segmentation of the vocal cords, the glottis ROI, and aspirated boluses was used to calculate the model’s segmentation performance (Dice score). Given two binary masks $X=\left(x_{ij}\right)$ and $Y=\left(y_{ij}\right)$, the Dice score is defined as
$$Dice\left(X,Y\right)=\frac{2\sum_{i,j}x_{ij}y_{ij}}{\sum_{i,j}x_{ij}+\sum_{i,j}y_{ij}}$$

To assess the model’s capability of correctly identifying frames where the glottis was not visible, the number of pixels falsely segmented as glottis ROI were calculated on all frames labeled as not containing the glottis. A confusion matrix was calculated to make the aspiration detection capabilities of the AI assessable. The detection performance was represented by its precision (positive predictive value: how many findings were actually aspirations) and recall (sensitivity: how many of the aspirations were found) metrics [58]. Based on these metrics, the *F*1 score as the harmonic mean of recall and precision was also calculated, to rate the AI performance between 0 and 1 in a single metric [59]. Given the number of true-positive (*TP*), false-positive (*FP*) and false-negative (*FN*) predictions of aspirations, the metrics are defined as:$$Precision=\frac{TP}{TP+FP}$$
$$Recall=\frac{TP}{TP+FN}$$
$$F1=\frac{2}{Recall^{-1}+Precision^{-1}}=\frac{2TP}{2TP+FP+FN}$$

Spearman’s rho correlations were calculated for the overlap of AI-segmented boluses with the size of the reference segmentation in order to investigate whether larger entities/more pixels could be detected more easily. The calculation of the metrics and the correlation analysis were performed using the Python packages scikit-learn v0.24.2 [60] and SciPy v1.5.2 [61].

### 2.4. Timeline for Interpretation of the Model Outcome

The XAI concept of our approach was based on a human-centered design of the model output. In order to enable full perceptive interpretability of the model outcome by a post hoc analysis that relies on the expert knowledge of the diagnostician, we implemented the concept of identifying meaningful or key frames in sequences [8,9]. This became possible because we automated the video analysis and applied the CNN to an entire video to generate a new video in which all AI-based segmentations and detections of aspirations were drawn into all frames of the video sequence. Furthermore, on a separate screen window, a timeline was generated that plotted a curve displaying the number of pixels for the segmentation tasks and the detected aspiration candidates. This provided the examiner with an overview across the complete video captured at one glance, resulting in a human-in-the-loop process. In other words, one could then look by scrolling at the time points where aspiration was detected on several consecutive frames to decide about the correctness of the AI detection. This human–computer interaction guarantees that the demands of the EU GDPR [44] are met and provides transparency to the user.

## 3. Results

In order to provide transparency for the development and explainability for the system process, the general distribution of the videos in the different datasets will be shown, and then the number of annotated frames is presented. After that, first general results for the AI performance are outlined before going into more detail on the segmentation and detection results. Finally, the XAI approach will be demonstrated.

### 3.1. Video Distribution across Data Sets and Annotated Frames

Ninety-two videos were included and were split into disjunct sets for training (77.2%, 71), validation during training (6.5%, 6), and final testing (16.3%, 15). Among the 50 videos with aspirations, the distribution of bolus types included slurry (21), saliva (18), and liquids (11). During preparation, 1330 frames were segmented and 2895 frames were labeled as not showing the glottis. Table 1 shows their distribution across the three data subsets for the development and evaluation of the AI.

### 3.2. AI Training

Figure 2 shows the learning process of the AI. The curves of the learning progress for glottis and vocal cord segmentation (Jaccard scores 1 and 2) rise steeply from the beginning, unlike the aspiration detection task (Jaccard score 4), where the AI does not learn until about 19,000 iterations. After the rise of Jaccard score 4 (aspiration detection), only the training loss curve, but not the validation loss curve, progresses to decline. The best model performance based on the mean Jaccard score is reached at 32,000 iterations, building the basis for the test run.

### 3.3. AI Performance

Boxplots for the Dice scores in Figure 3 show high values for the segmentation of the glottis and few lower values for the vocal cords across all data subsets, with median values of 0.94 and 0.85, respectively, on the test set.

The plot clearly shows some overfitting, despite early stopping, as the performance is, overall, higher on the training set than on the validation and test sets, particularly for aspiration segmentation. During training, the segmentation of aspiration achieves a median Dice score of 0.75 but drops to a median of 0.32 during validation and 0.13 during testing, accompanied by a large increase of the inter-quartile range. On frames that were labeled as not containing the glottis, the model detects false positive pixels in 5.6 %, 4.3 %, and 9.4 % of frames for training, validation, and test sets, respectively.

For aspiration detection, a confusion matrix was also calculated (Figure 4) to determine the performance of the AI. Whenever aspiration was segmented and the Dice overlap with the reference was greater than 0, the frame was counted as true-positive detection. For the training data, a very good result could be obtained for true-negative outcomes and a good result could be obtained for true-positive outcomes. The achieved value for false positives is in the lower range, which is the range for false negatives as well.

Additionally, we calculated the resulting values for precision, recall, and *F*1 score (Table 2). On the training set, the precision was very high (0.955), meaning that most detections were indeed aspirations. This dropped to 0.5 during validation and 0.706 during testing. Among all annotated aspirations, the AI detected 90% during training but only 59% during validation and 57% during testing. The harmonic mean of both metrics (*F*1 score) also dropped from 0.925 to 0.541 and 0.632 for training versus validation and testing, respectively. As for the segmentation, this also showed the overfitting on the training set.

Selected video frames in Figure 5 demonstrate this heterogeneity of results. When the glottis is well visible, the segmentation of the glottis and the vocal cords is very precise (Figure 5a–c), but may be less robust when the glottis is only partially visible or near the image edge (Figure 5d). Aspirations can be detected in the correct location (Figure 5d–e), but can also be overlooked (Figure 5f) or falsely detected—for example, due to light reflections (Figure 5g). In addition, an ROI segmentation can appear even though the relevant anatomical structure is not visible/present within the respective frame (Figure 5h, piriform recess). Despite a correct detection of the aspiration itself, as in (Figure 5d–e), the segmentation itself may be imprecise, leading to low Dice score values.

In order to investigate whether the bolus segmentation might depend on the amount of aspirated bolus (e.g., the more aspirate, the easier it may be detected), as visualized in Figure 6, we calculated Spearman’s rho correlations for the overlap of reference and AI segmentation (Dice score) with the size of aspiration (number of pixels segmented in reference). The correlation decreases from training (r = 0.62, *p* = 0) to validation (r = 0.43, *p* = 0.08) and testing (r = 0.37, *p* = 0.003).

### 3.4. Interpretability by Identifying Meaningful Frames

As a means for post hoc interpretation of the model outcome by the examiner, we implemented a concept of identifying meaningful frames in sequences. Therefore, an automated video analysis applies the CNN to an entire video to create a new video in which all AI-based segmentations and detections of aspirations are drawn into all frames of the video sequence (Figure 7), serving as a first visual aid for key frames. The unmarked video can be seen in parallel. As a second visual aid, on a separate screen window a timeline is generated that plots a curve displaying the number of pixels for the segmentation tasks and the detected aspiration candidates. It also features a further zoom window. Hence, the examiner is provided with an overview across the complete video, captured at one glance, and can scroll to meaningful frames for diagnostic purposes.

In the given example (Figure 7), slurry parts of yoghurt and saliva become aspirated and reside above the first cartilage of the trachea (membrana cricothyroidea). The AI detects the part next to the vocal cords as an aspiration.

## 4. Discussion

The discussion will first focus on the XAI aspects of our approach; then, the model accuracy will be mooted.

Due to the human-in-the-loop process and the HCI, our XAI can be considered a “hybrid” concept that combines data- and knowledge-driven approaches, as well as white- and black-box modeling approaches. Our attempt provides full post hoc human-based perceptive interpretability of the model outcome by the examiner. Hence, our concept of identifying meaningful frames by adding visual aids adds a further example to the notion of key frame identification as XAI approaches [8,9]. The user can decide, if the AI explanation is suitable, and based on that the further course for the patient can be planned (e.g., oral feeding is possible). Therefore, the final explanation provided by the system is effective and acceptable. This goes beyond most existing approaches for this task (except the VFSS approach [25]), because they only provide predictions or classifications without providing proper interpretable information for the diagnostician [28,35,37,42,43,62,63]. As this lack of transparency conflicts with EU GDPR, which prohibits decisions based solely on automated processing [44,45], a subsequent FEES or VFSS would become necessary, in any event, before critical decisions—such as abstinence from food, insertion of a nasogastric tube, or even re-intubation and tracheotomy—could be made. Furthermore, regarding interpretability, our concept of meaningful frames not only enables interpreting the model output for diagnostic purposes, but also facilitates the ongoing quality and performance assessment of the model compared to a patient-level black-box prediction, helping to further develop our decision-support system. Additionally, since in current FEES practice a retrospective video analysis may already be preferable [18] but is very time-consuming, our interpretable model output is an appropriate tool for focusing on relevant meaningful frames, instead of viewing the whole video again.

Regarding the accuracy of the segmentation of anatomical structures, we obtained very satisfying results that were similar to comparable work in the field [46,47,48]. Despite using only selected frames for training and not full videos, we achieved a false-positive rate for glottis segmentations of only 5% on frames labeled as not containing the ROI, allowing for the processing of full videos. We expect to be able to further reduce this rate by labeling more negative example frames, which is a relatively fast annotation operation, as no segmentation is required. Taking into account information from consecutive frames (e.g., using a recurrent network architecture) would likely help to further reduce spatiotemporal noise in the predictions.

Hence, we conclude that the general requirement for the second step, the AI-based detection of aspiration, was fulfilled. To be of use for the clinical workflow, both high recall (i.e., identification of true aspiration events) and high precision (i.e., not too many false positives) are desirable. In our preliminary study, we achieved satisfying precision during training (0.955) and testing (0.706), but slightly lower recall during training (0.925) and testing (0.571), meaning that a large amount is still overlooked. A trade-off between precision and recall is typical for detection algorithms; therefore, we might increase recall at the cost of lower precision. Newly emerging false-positive detections might be eliminated in a post-processing step. The segmentation accuracy of the detected aspiration was satisfactory during training (Dice score 0.75) but still needed to be improved during validation (0.32) and testing (0.13), and was accompanied by a large increase in the inter-quartile range. Overall, the decline in detection and segmentation performance from training to validation and testing was unsatisfying. In a qualitative analysis, we looked at samples of mispredictions to evaluate whether the type of bolus (slurry, saliva, liquid) played a role, especially since we had no equal distribution for them in the training data; however, we were not able to identify such a contributing factor. When considering the size of the aspiration as a potential explanation for its detectability within a frame with known aspiration (i.e., a segmentation), we saw a strong correlation of the true bolus size with the Dice score in the training data, but it was not as high during validation and testing. While the Dice score itself is known to correlate with the area-to-contour ratio of a 2D object, this still indicates that other factors, in addition to bolus size, may impact the segmentation accuracy during testing—for example, changed lighting conditions. This limited performance can already be seen in the loss and validation plot of the training process (Figure 2), where the AI shows signs of overfitting to training data.

Hence, the currently trained AI lacks sufficient generalization for aspiration detection but not for segmentation of vocal cords and glottis ROI. Regarding the aspiration detection task, the current model performance might be comparable to that of an untrained human examiner [21]. Hence, at present, our model does not lead to better results than comparable non-endoscopic/non-radiologic approaches [28,33,35,37,62,63]; but in clear contrast to them, our model outcomes, as well as the false positives and negatives, are fully interpretable and can therefore be corrected by an experienced examiner. This becomes particularly easy, because the examiner can perform the correct assignment by jumping to the respective point in the timeline of the video sequence.

Because explainability forms a crucial aspect of XAI but needs to be accompanied by a profound model performance, and because only this concurrence will lead to the acceptance of our developed system, we are at present in a re-evaluation process to understand the limitations of our model for aspiration detection. As explanations for this particularly unsatisfying model performance at present, we have already identified various limiting aspects that can be specifically addressed. First, with 4225 annotated (thereof 1330 segmented) frames, we have only achieved a basic sample for the training. Furthermore, we did not achieve a homogeneous distribution of annotated frames regarding the different subtypes of aspiration (i.e., slurry, saliva, liquids); this was only provided for the samples with and without aspiration (50 vs. 42). Therefore, regarding the training data, there were far more frames in which the ROI appeared than in which aspirations occurred. We did apply a sampling strategy to account for part of the imbalance; however, we did not fully optimize the ratio of frames with and without aspiration, or not showing the ROI at all, in a hyperparameter tuning step. In the future, we will, therefore, include more patient videos, annotate significantly more frames (especially more frames with aspirations), and, in addition, apply more data augmentation techniques to strengthen the robustness. Moreover, we have currently processed the videos in a pure 2D approach, analyzing the video frame by frame. The 2D approach for training and prediction was chosen on the basis of our sparsely labeled training set, in which only a few frames per video were manually annotated and could be directly used for supervised training. To further strengthen the aspiration detection, we will consider a 2D + T approach—for example, using recurrent neural networks, which take a temporal sequence of frames into account. To achieve this, we need to explore strategies for combining labeled and unlabeled frames in the training. Additionally, we want to implement an online augmented reality approach to highlight moments of potential aspiration as detected by the AI in a separate small window, while the endoscopic procedure can continue. This would enable real-time verification, possibly with an adjustment of the FEES procedure (e.g., retesting a certain type of bolus).

As a further future goal, as well as a general idea for other research groups engaged in the development of XAI systems in the field of dysphagia diagnostics, we propose to implement a feedback system, especially for corrections and negative feedback information, that can be provided by the domain experts. In such an active learning scenario, the algorithm itself could suggest frames in which it is not clear whether aspiration was detected or not and ask for feedback. This would enable continuous training or planned re-training in certain intervals to enhance the model performance, while at the same time reducing annotation effort when compared with that of an undirected approach. Moreover, when gaining research partners who are in possession of a reasonable amount of narrow-band imaging videos showing aspirations [23,24], this could also be used to further facilitate the AI-based detection. Additionally, we could implement other XAI concepts, such as a combination of frame-wise classification for aspiration detection and XAI methods such as GradCAM or Saliency maps [3]. We could compare the output of these methods to the proposed segmentation to evaluate their usefulness as visual aids. Taken together, and despite the discussed limitations of the current model’s state, our novel concept of AI-based detection of aspiration during video-endoscopy with visual aids in meaningful frames makes it possible to interpret the model outcome. With the proposed XAI approach, the AI segmentation and the pixel-wise classification as an aspiration can be verified, thereby providing proper interpretable information for the diagnostician to understand why subjects were classified, and beyond that, enabling the identification of misclassifications. This substantially reduces the black-box character of the machine-learning model. Therefore, our current attempt is an important step in making the identification of meaningful frames an XAI approach that will become more applicable in clinical contexts.

## 5. Conclusions

For the first time, we have introduced an XAI that has been trained to detect aspiration in endoscopic swallowing videos. While detection performance needs to be optimized significantly in future studies, our architecture resulted in a final model that explains its assessment by locating specific video frames with relevant aspiration events and by highlighting the suspected bolus in situ as a meaningful sequence. Hence, in contrast to existing machine-learning tools for aspiration detection, the AI decision in our framework is verifiable, interpretable, and, thus, accountable for clinical users. During the next development steps, the interaction with dysphagia experts will continuously improve the outcome.

After the implementation of this tool in FEES software, it will aid endoscopists in improving accuracy (thereby potentially saving lives), shorten the duration of the administration, and save overall costs, as positive contributions to healthcare.

## Figures and Tables

**Figure 1 sensors-22-09468-f001:**
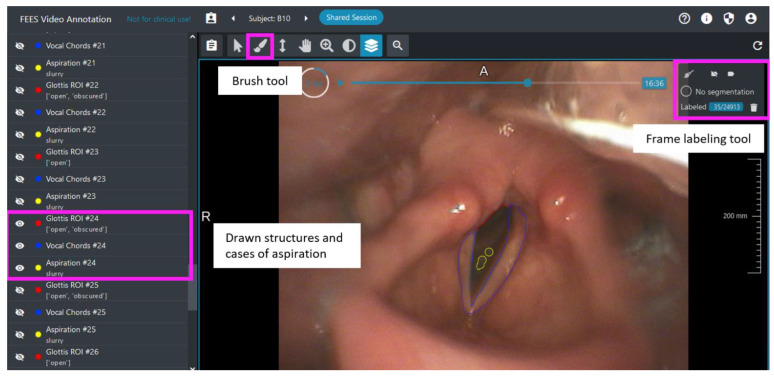
Annotation example with drawn structures of vocal cords (red), glottis (blue) and cases of aspiration (yellow) using the frame-labeling tool (screenshot).

**Figure 2 sensors-22-09468-f002:**
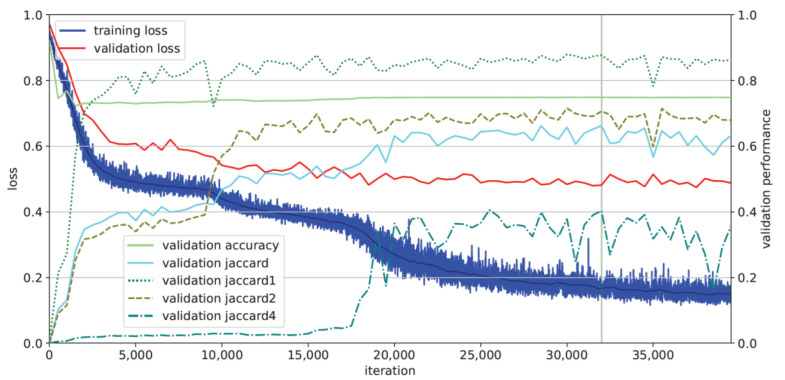
Loss curves of training (blue) and validation (red), as well as validation Jaccard scores (turquoise = mean of all; 1/dotted = segmentation of glottis; 2/dashed = segmentation of vocal cords; 4/dashed and dotted = detection of aspiration), show overlaps with the references. The vertical line shows the moment of the optimally working model.

**Figure 3 sensors-22-09468-f003:**
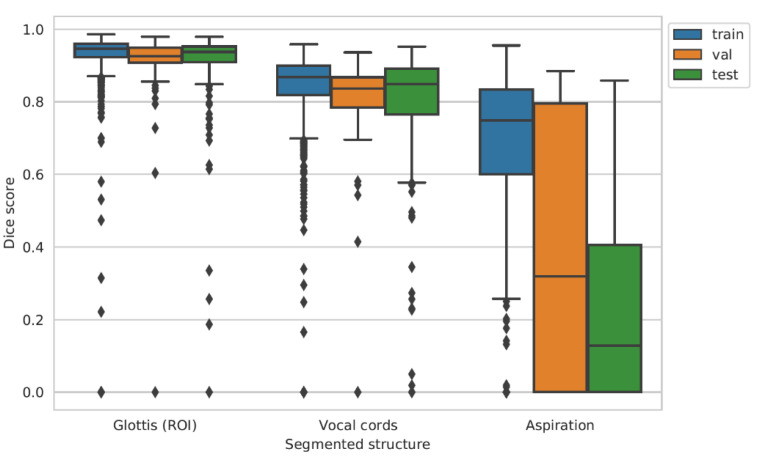
Boxplots of the Dice scores for all tasks (segmentation of the glottis, vocal cords, and detection of aspiration) for training, validation, and testing. Diamonds denote outliers that deviate more than 1.5 times the inter-quartile range from the third quartile.

**Figure 4 sensors-22-09468-f004:**
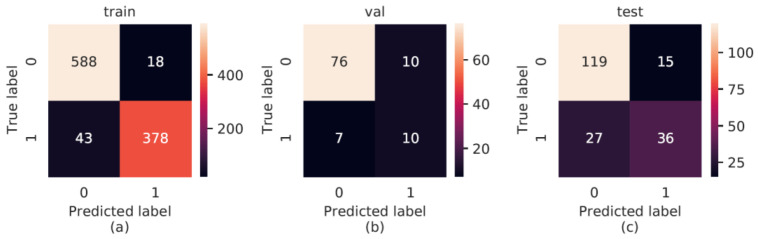
Confusion matrices with predicted and true labels (0 = negative/1 = positive) for aspiration detection for training (**a**), validation (**b**), and testing (**c**), with heat-map scales for result interpretation (right in each case).

**Figure 5 sensors-22-09468-f005:**
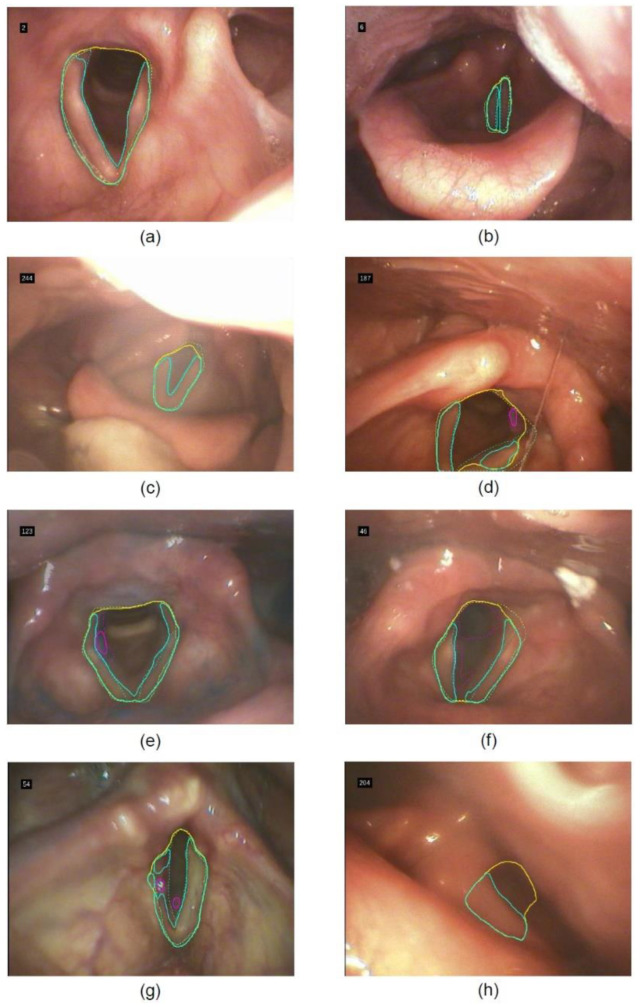
Examples of segmentation results in the test set across different videos: (**a**–**c**) high overlap between references (dotted) and AI-based (drawn through) segmentation in different states (open, closed) and light conditions, (**d**) segmentation errors of partially visible glottis close to the image edge, (**e**) correct detection of aspiration, (**f**) missed detection of aspiration, (**g**) false-positive detection of aspiration, (**h**) false-positive segmentation of glottis and vocal cords on frame without visible glottis. Solid lines denote the automatic segmentation (yellow: glottis ROI, cyan: vocal cords, magenta: detected aspiration), dotted lines the reference segmentation. Numbers in the upper left corner denote the index of the frame in the test set.

**Figure 6 sensors-22-09468-f006:**
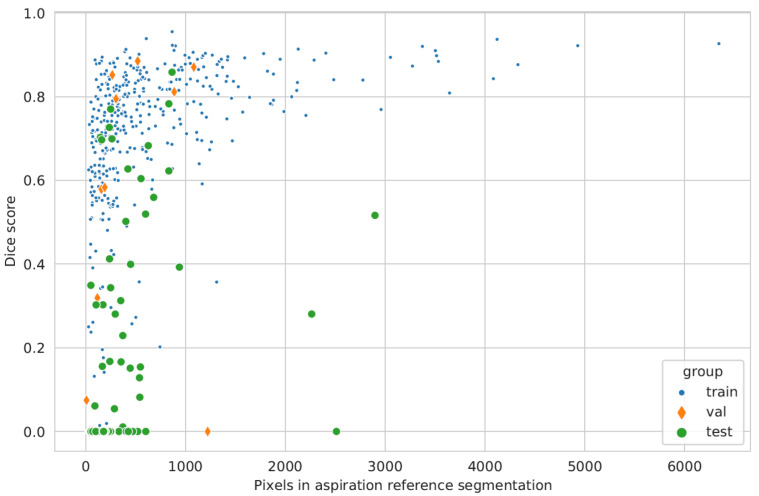
Scatterplot of the overlap between the number of pixels for the aspiration in the reference annotation (x-axis) and the Dice score for the overlap of AI detection (y-axis) and reference during training, validation, and testing.

**Figure 7 sensors-22-09468-f007:**
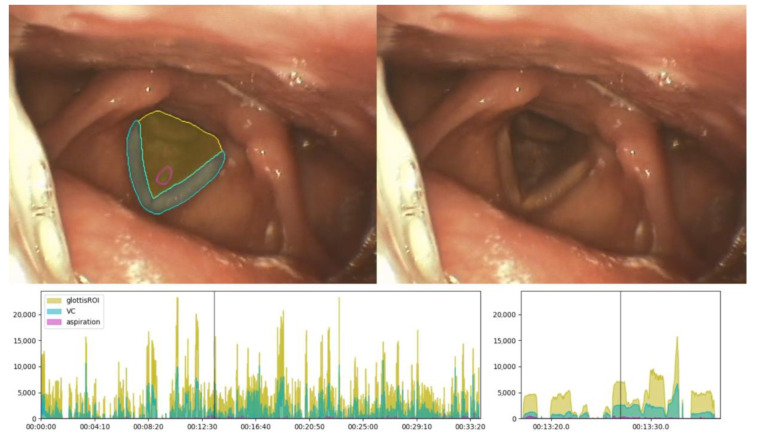
Visual aids to find meaningful frames for interpretation of model output. Screenshot of AI-based segmentation and detection of aspiration results (upper left), respectively, and normal view (upper right). Timeline and timeline zoom with a curve displaying the number of pixels for the segmentation tasks and the detected aspiration (below). Vertical line indicates the point in time.

**Table 1 sensors-22-09468-t001:** Distribution of annotated frames across the different data sets.

AI Data Subset	Segmented Frames	Frames with Aspiration	Frames Not Showing Glottis
training	1029	424	2220
validation	103	17	186
test	199	63	489

**Table 2 sensors-22-09468-t002:** Metrics for aspiration detection for all data sets.

Metrics	Training	Validation	Test
Precision	0.955	0.500	0.706
Recall	0.898	0.588	0.571
F1 score	0.925	0.541	0.632

## Data Availability

The data are not publicly available due to restrictions on the use of clinical patient data. Furthermore, a public accessibility is not covered by the given ethics vote. In this, data sharing is limited to the research partner (MEVIS).

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
