# Peer review of "AI-Based Detection of Aspiration for Video-Endoscopy with Visual Aids in Meaningful Frames to Interpret the Model Outcome"

_sensors, 2022, doi:10.3390/s22239468_

Round 1

Reviewer 1 Report

Quite a large team of Authors undertook to find a solution to an extremely difficult problem, which is AI-based detection of aspiration for video-endoscopy with visual aids in meaningful frames to interpret the model outcome. In my opinion, the task was done perfectly. The manuscript reads fluently, the content is consistent, and the rich bibliography allows you to supplement your knowledge if necessary. The described XAI model has been trained to detect aspiration in endoscopic swallowing videos. Besides, it explains its assessment by locating specific video frames with relevant aspiration events. Moreover, the model distinguishes the suspected bolus in situ as a meaningful sequence. Thus, these difficult diagnostic decisions are verifiable, interpretable, and thus acceptable for clinical users. Of course, then the interaction with the dysphagia experts can improve the outcome. The undoubted advantages of the recommended software are aid endoscopists to improve accuracy, shorten the duration of the administration, and safe costs. It is worth emphasizing that disorders of swallowing are a relevant problem across various etiologies and all sectors of healthcare provision. Generally speaking, oropharyngeal dysphagia is extremely dangerous to human life, even in such trivial cases as aspiration of boluses and saliva, when material passes the vocal cords and enters the airways.

Admittedly the high potential application is the CNN for aspiration detection of VFSS videos with an accuracy of AUC of 1.00, but it is limited in clinical use. In the proposed model, the AI ​​learns the segmentation of relevant anatomical structures like the vocal cords and the glottis. Simultaneously, the AI ​​is trained to detect bolus that passes the glottis and becomes aspirated into the airways. This interpretable architecture results in a final model that explains its assessment by locating specific video frames with relevant aspiration events and by highlighting the glottis, vocal cords, and suspected bolus in situ as visual aids in meaningful frames.

In my work, I encountered some weaker passages worth discussing:

1. Automated the video analysis by the CNN to generate a new video in which all AI-based segmentations and detections of aspirations are drawn into all frames of the video sequence was applied. Why was the CNN not compared to another LSTM network which is considered an effective tool for video clip analysis?

2. The math formulas are missing from the manuscript. Even if we write about medical issues, it is worth defining basic relationships, such as F1 score. We cannot assume that the Reader remembers all the concepts, especially since we are writing about XAI.

3. The very nice writing style of the work is slightly disturbed by the aesthetics of Figure 3, in which the size of the font describing the axes is approx. 50% larger than that used in the manuscript.

Of course, the above comments do not negatively affect the very high rating of the article, but taking them into account will undoubtedly increase its quality.

Reviewer 2 Report

Dear authors,

I found your paper interesting and well-written. Despite thorough checking, I could not find any major problems. The topic is fresh and interesting, and the scope of the work clearly presented and well-organized. In my opinion, the language is fine and I could not find any big issues. The only thing which lowered the reception of the paper was the quality of the figures. Please make them more readable (e.g. fig 1 is blurred and too small, the font on fig 2 is too small, and the figure itself would benefit from being bigger).
